# Intelligent Identification of Early Esophageal Cancer by Band-Selective Hyperspectral Imaging

**DOI:** 10.3390/cancers14174292

**Published:** 2022-09-01

**Authors:** Tsung-Jung Tsai, Arvind Mukundan, Yu-Sheng Chi, Yu-Ming Tsao, Yao-Kuang Wang, Tsung-Hsien Chen, I-Chen Wu, Chien-Wei Huang, Hsiang-Chen Wang

**Affiliations:** 1Division of Gastroenterology and Hepatology, Department of Internal Medicine, Ditmanson Medical Foundation Chiayi Christian Hospital, Chia Yi City 60002, Taiwan; 2Department of Mechanical Engineering, Advanced Institute of Manufacturing with High Tech Innovations (AIM-HI) and Center for Innovative Research on Aging Society (CIRAS), National Chung Cheng University, 168, University Rd., Min Hsiung, Chia Yi City 62102, Taiwan; 3Division of Gastroenterology, Department of Internal Medicine, Kaohsiung Medical University Hospital, Kaohsiung Medical University, No. 100, Tzyou 1st Rd., Sanmin Dist., Kaohsiung City 80756, Taiwan; 4Department of Medicine, Faculty of Medicine, College of Medicine, Kaohsiung Medical University, No. 100, Tzyou 1st Rd., Sanmin Dist., Kaohsiung City 80756, Taiwan; 5Graduate Institute of Clinical Medicine, College of Medicine, Kaohsiung Medical University, No. 100, Tzyou 1st Rd., Sanmin Dist., Kaohsiung City 80756, Taiwan; 6Department of Gastroenterology, Kaohsiung Armed Forces General Hospital, 2, Zhongzheng 1st Rd., Lingya District, Kaohsiung City 80284, Taiwan; 7Department of Nursing, Tajen University, 20, Weixin Rd., Yanpu Township, Pingtung County 90741, Taiwan; 8Director of Technology Development, Hitspectra Intelligent Technology Co., Ltd., 4F., No. 2, Fuxing 4th Rd., Qianzhen Dist., Kaohsiung City 80661, Taiwan

**Keywords:** hyperspectral imaging, single-shot multi-box detector, white-light endoscopy, narrow-band endoscopy, band selection, dysplasia

## Abstract

**Simple Summary:**

Early esophageal cancer detection is crucial for patient survival; however, even skilled endoscopists find it challenging to identify the cancer cells in the early stages. In order to categorize and identify esophageal cancer using a single shot multi-box detector, this research presents a novel approach integrating hyperspectral imaging by band selection and a deep learning diagnostic model. Based on the pathological characteristics of esophageal cancer, the pictures were categorized into three stages: normal, dysplasia, and squamous cell carcinoma. The findings revealed that mAP in WLIs, NBIs, and HSI pictures each achieved 80%, 85%, and 84%, respectively. The findings of this investigation demonstrated that HSI contains a greater number of spectral characteristics than white-light imaging, which increases accuracy by roughly 5% and complies with NBI predictions.

**Abstract:**

In this study, the combination of hyperspectral imaging (HSI) technology and band selection was coupled with color reproduction. The white-light images (WLIs) were simulated as narrow-band endoscopic images (NBIs). As a result, the blood vessel features in the endoscopic image became more noticeable, and the prediction performance was improved. In addition, a single-shot multi-box detector model for predicting the stage and location of esophageal cancer was developed to evaluate the results. A total of 1780 esophageal cancer images, including 845 WLIs and 935 NBIs, were used in this study. The images were divided into three stages based on the pathological features of esophageal cancer: normal, dysplasia, and squamous cell carcinoma. The results showed that the mean average precision (mAP) reached 80% in WLIs, 85% in NBIs, and 84% in HSI images. This study′s results showed that HSI has more spectral features than white-light imagery, and it improves accuracy by about 5% and matches the results of NBI predictions.

## 1. Introduction

Noncommunicable diseases account for 70% of the world’s total deaths [1,2,3]. Among them, malignant tumors, also known as cancers, are the second leading cause of death in the world according to the latest statistics in 2020 [4,5,6]. Esophageal cancer has the sixth highest fatality rate [7,8,9]. Nearly 90% of esophageal carcinogenesis occurs in the mucosal layer, which is composed of multiple layers of squamous epithelial cells; thus, esophageal carcinoma can be divided into two types based on its cancerous location [10]. The first type is squamous cell carcinoma (SCC), and the other type is called adenocarcinoma (AC) [11,12]. AC is more common in America and Europe, whereas SCC is more common in Asia, Japan, and Taiwan [13,14]. Esophageal cancer is difficult to identify in its early stages given the lack of evident symptoms [15,16]. By the time esophageal cancer is detected, it is already in the second or third stages. 

Although the incidence of esophageal cancer is comparatively lesser than that of other cancer types, the average survival rate is less than 10%, making it a fatal disease [17]. However, if esophageal cancer is detected in its early stages, the five-year survival rate is 90%; however, if it is detected in the later stages, the five-year survival rate drops below 20% [18,19]. Therefore, detecting esophageal cancer in its early stage is vital to increasing the five-year survival rate. One of the methods of detecting esophageal cancer is through a biosensor; however, most biosensors cannot detect cancer at the early stages [20,21,22,23].

In recent years, artificial intelligence (AI) has been extensively used in medical imaging to detect cancer lesions, and most of the methods exhibit a good performance [24,25,26,27,28,29]. Horie et al. used convolutional neural networks (CNNs) to train and predict esophageal cancer, and the sensitivity value was as high as 98% [30]. However, the dataset used in this study was considerably small. In 2020, Albert et al. compared the results of a model to detect early Barret′s esophagus with the results diagnosed by physicians, and they confirmed that the deep learning model outperformed the physicians [31]. At present, the related studies on predicting the spectrum of esophageal cancer using hyperspectral imaging (HSI) combined with AI are limited. Most of them use spectrometers for actual measurements. Maktabi et al. collected images of cancer lesions during surgery and measured the spectrum [32]. However, the patient selection in this study was biased toward patients who were already sick.

HSI has previously been used in numerous classifications fields, such as agriculture [33], astronomy [34], military [35], biosensors [36], air pollution detection [37,38], remote sensing [39], dental imaging [40], environment monitoring [41], satellite photography [42], cancer detection [43], forestry monitoring [44], food security [45], natural resource surveying [46], vegetation observation [47], and geological mapping [48]. The advantage of HSI lies in its excellent resolution, and with a minimum spectral resolution of less than 10 nm, it can be used to obtain more information than RGB images [49]. In addition, the spectral features of different substances are unique, which can provide better identification capability, making the model more accurate for identifying features [50,51]. However, such a large amount of spectrum information greatly increases the dimension of data. If the amount of data cannot meet the high-dimensional requirements, the accuracy will reduce. Therefore, the method of reducing the dimension is very important. Spectral redundancy must be removed while retaining important characteristics. The advantage of band selection is that a specific band or subset can be directly selected, thereby preserving most of the spectral information [52].

In this study, HSI (Smart Spectrum Cloud, Hitspectra Intelligent Technology Co., Ltd., Kaohsiung City, Taiwan) was combined with band selection and used to convert esophageal cancer images into spectral images to reduce the dimensions, and color reproduction was used to convert white-light images (WLIs) into narrowband endoscopic HSI images (NBI). Finally, deep learning was used to classify the images into three categories: normal, dysplasia, and SCC.

## 2. Materials and Methods

In this study, a total of 1780 esophageal cancer images, including 845 WLIs and 935 NBIs, were used. These images were divided into three categories: normal, dysplasia, and SCC. The WLI category comprised 470 normal, 156 dysplasia, and 219 SCC images, and the NBI category consisted of 425 normal, 290 dysplasia, and 220 SCC images. The images that were blurry, full of mucus, and contained a number of bubbles for the identification of lesions were excluded. 

First, the endoscopic images were cut and zoomed in to remove the patient information and unnecessary noise from the images. Then, the doctors of Kaohsiung Medical University marked the processed images into three categories and exported the data of the marked images into the mark format required by the model. This dataset was amplified by rotating the images at different angles and cropping them. The spectral information from WLI was obtained in the visible-light band (380–780 nm, 401 bands) using the HSI conversion process (Figure 1). The 24 color patches were used as the benchmark target. After obtaining the spectral information and images of the 24 color blocks with a spectrometer and an endoscope camera, respectively, the two datasets were converted into numerous color spaces to obtain their conversion matrix, and the RGB images were converted into spectral information.

Two wavelengths, namely 415 and 540 nm, were selected to obtain the spectral information converted from the image. These wavelengths were specifically selected because of the different types of light in the images. The longer the wavelength, the deeper the penetration [53]. The red light is absorbed differently by heme in blood vessels based on the difference in depth [54]. As a result, the microvessels in the superficial mucosa tissue will be brown, and blood vessels in the submucosal tissue will be green, thereby creating a strong sense of hierarchy that is advantageous to identifying lesions of the mucosal tissue. Figure 2 shows the comparison of the WLIs and HSI images. The red color of blood vessels in the lesion in Figure 2a and those in Figure 2b were converted to purplish red color, which improved the contrast with the background, allowing doctors to more easily observe and diagnose cancer in its early stages.

Single-shot multi-box detector (SSD) is a single-shot multi-category target detector that is constructed based on CNNs [55] (see Appendix A for the detailed explanation of SSD). The SSD used in this study had a detection architecture based on the VGG-16 network. VGG-16 is a deep learning architecture with 16 hidden layers consisting of 13 convolutional layers and 3 fully connected layers (see Appendix A for the detailed explanation of the model used in this study) [56]. The dataset was divided into three categories, namely WLI, NBI, and HSI endoscopic images, and allotted to a training set and a test set. The training set contained 601 WLI and 711 NBI endoscopic images. The test set comprised 244 WLIs and 224 NBIs. The HSI endoscopy image was the same as the WLI endoscopy image. Finally, the three test sets were inputted to the SSD model for training, and the results were evaluated with the test set. 

The results of the prediction models were presented in two ways: visualization of the prediction frame and ground truth, with confidence in the prediction. The degree of coincidence between the prediction results and the marked position was compared with the degree of influence of the prediction model on the prediction results. Next, the prediction performance of the models was analyzed with several evaluation indicators, such as sensitivity, precision, F1-score, kappa value, and average precision (AP). The sensitivity indicates how well the model can detect symptoms of esophageal cancer. The accuracy value indicates the proportion of esophageal cancer and actual cancer symptoms in the model′s diagnosis. F1-score is a harmonic mean, and it can be used as a rough indicator of the model performance. Kappa value can be used to evaluate the consistency between prediction and pathological analysis results to evaluate the feasibility of prediction tools. Its value is between −1 and 1 and often has a threshold of 0.6. AP is a commonly used evaluation index for object detection, and the overall sensitivity and accuracy are used to quantify the overall performance of a prediction model.

## 3. Results

Figure 3 shows the sketch of the predictions from the three models developed. Table 1 shows the results of the prediction model based on the evaluation criteria. The precision, accuracy, AP, and kappa values were expressed in the order of normal, dysplasia, and SCC. The accuracies were 84.5%, 84.9%, and 87% for the WLI prediction model; 87.6%, 84%, and 87.4% for the NBI prediction model; and 90.9%, 89.7%, and 89.8% for the HSI prediction model. The sensitivity values were 69.3%, 81.6%, and 81% for the WLI prediction model; 79.2%, 69.7%, and 80.8% for the NBI prediction model; and 71.2%, 92.1%, and 85.6% for the HSI prediction model. The AP values were 75.3%, 81.2%, and 85% for the WLI prediction model; 84.5%, 84.2%, and 86.7% for the NBI prediction model; and 78.9%, 83.6%, and 88.5% for the HSI prediction model. The kappa values for WLI, NBI, and HSI were 0.60, 0.653, and 0.665, respectively.

The NBI model showed a good performance in the SCC category and better prediction of the normal category compared with the other two models. This finding may be due to the higher consistency of the blood vessel features in normal images, which is conducive to the prediction of the box selection, compared with those in the other two models. In addition, the dataset showed a profound influence on the prediction results. Compared with the WLI, the accuracy and sensitivity of the HSI model have increased greatly. Figure 4 presents a schematic of the comparison of the prediction results of the WLI and HSI models. The highlighted blood vessel features of the HSI images resulted in similar prediction findings and ground truth box, thereby increasing the confidence level. The indicators in the dysplasia category of the WLI model increased significantly, and the gap with other evaluation indicators widened, improving the performance of the dysplasia category compared with that of the SCC category.

## 4. Discussion

Given the three data categories in the HSI model, except for the normal category, the other two exhibited a good performance. Although HSI images highlighted the blood vessels in the WLIs, comparing the original WLI with NBI, several blood vessel features were relatively lacking, resulting in incomplete blood vessel features after conversion into a spectral image. Thus, NBI performed better in the normal category. In terms of mAP, the HSI method did not obtain the highest results. However, the average mAP was only low because of the normal category, where the NBI method exhibited the highest performance. If only the other two categories were considered, namely dysplasia and SCC, HSI would have had a better mAP. The HSI method also presented the highest kappa value. All kappa values exceeded the threshold of 0.6, which indicates that the model in this study is feasible for the application in esophageal cancer detection. In addition, the performance of WLI was relatively better than that of NBI in the dysplasia category. However, logically, the results should be the opposite due to the difference in the images of the two datasets. Therefore, the dataset still needs to be improved. 

This research proved that the HSI images converted by hyperspectral technology and band selection had an upward trend for all indicators, which also meant that the prediction results improved significantly after the blood vessel features in the WLIs were strengthened. Moreover, comparing the performances of WLI and NBI, highlighting vascular features can improve the prediction performance. The research based on AI-related studies using HSI for early esophageal cancer is limited, and most of the studies use hyperspectral imagers to obtain the actual spectrum and apply different dimensionality reduction methods to reduce the dimension of the spectrum data before training. However, the most suitable dimensionality reduction method for HSI to detect esophageal cancer has not been found. In addition, even though the accuracy of the model has been improved significantly by HSI, the accuracy can be further improved by increasing the dataset. In the future, the same model needs to be trained on marginally different samples of the training data to reduce variance without any noticeable effect on bias. Furthermore, the exact machine learning model that is perfectly suitable with HSI needs to be established by comparing the performance markers of different machine learning models. The related research on the spectrum of esophageal cancer has not yielded clear results, such as whether special features can be used for the imaging of the gastrointestinal tract. In addition, the new method proposed in this study can provide the function of NBI imaging for newer endoscopes, such as capsule endoscopes.

## 5. Conclusions

This study provides a new method for simulating WLIs into NBIs. Through the combination of HSI and band selection, the characteristics of blood vessels in WLIs can be enhanced. This method improves the diagnostic performance for early cancer. For early esophageal cancer, the simulated HSIs will increase the accuracy by up to 2% when compared with the traditional WLI. The improvement represents about 2–4% specificity, whereas the sensitivity improves by 5–10%. Therefore, this study will reduce the number of instruments to create the NBI of the esophagus, thereby increasing the ease of entering the body and reducing patient discomfort. Therefore, if the technology of this study can be further developed and combined with new endoscopes, it can provide patients with a more comfortable inspection experience, improve the inspection rate, and possibly enable early detection, thus achieving the purpose of early treatment and reducing mortality.

## Figures and Tables

**Figure 1 cancers-14-04292-f001:**
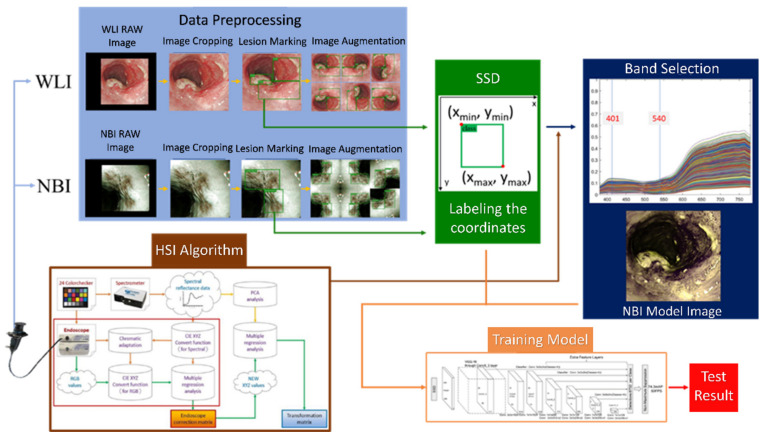
Flow chart of spectrum conversion construction using standard 24 color blocks (X-Rite Classic, 24 Color Checkers) as the common target object for spectrum conversion of the endoscope and spectrometer, converting the endoscopic images into 401 bands of the visible-light spectrum information (See Appendix A for the HSI algorithm).

**Figure 2 cancers-14-04292-f002:**
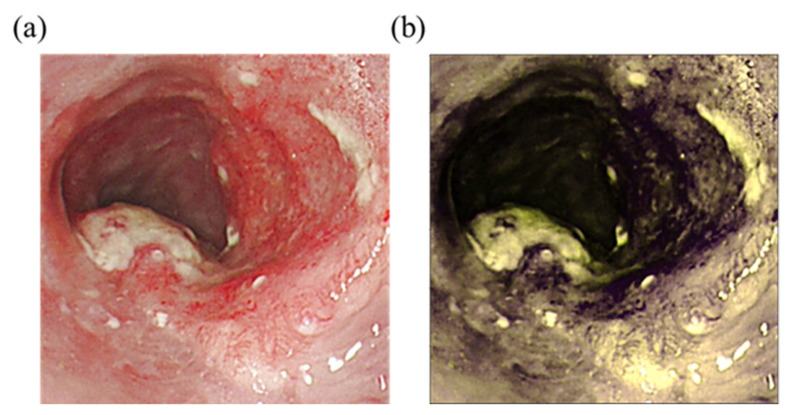
Comparison of WLIs and HSI images. (**a**) Original WLI; (**b**) HSI image of the NBI.

**Figure 3 cancers-14-04292-f003:**
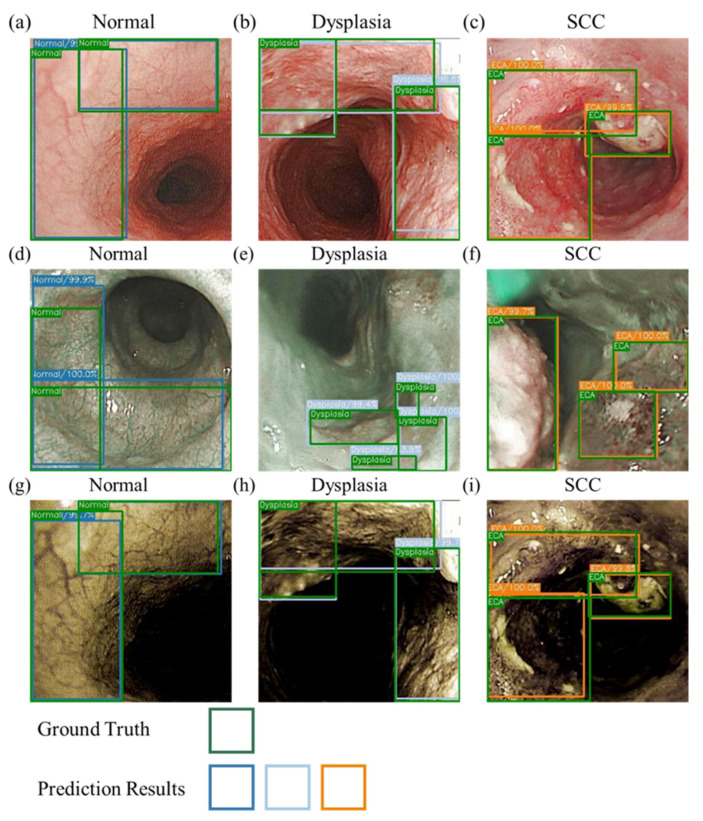
Schematic of the prediction results for WLIs, NBIs, and HSI images. (**a**–**c**) are the WLI prediction results. (**d**–**f**) are the NBI prediction results. (**g**–**i**) are the HSI prediction results. (**a**,**d**,**g**) are the normal period, and the green and blue boxes represent the real and predicted frames, respectively. (**b**,**e**,**h**) are the dysplasia period, and the green and gray boxes represent the real and predicted frames, respectively. (**c**,**f**,**i**) are the SCC period, and the green and orange boxes represent the ground truth box-predicted boxes, respectively.

**Figure 4 cancers-14-04292-f004:**
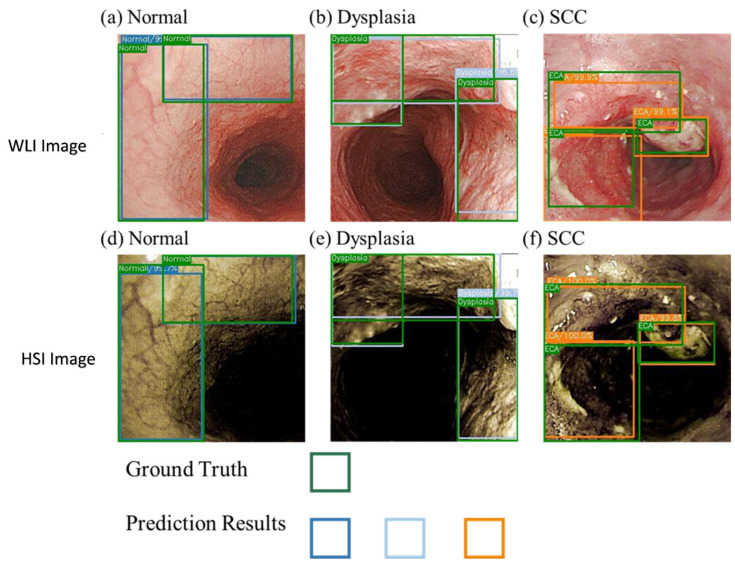
Comparison of the WLI and the HSI models.

**Table 1 cancers-14-04292-t001:** Predicted performance results of the three developed models.

WLI	Accuracy (%)	Precision (%)	Sensitivity (%)	F1 Score (%)	AP (%)	Kappa
Normal	78.3 (505/645)	84.5 (224/265)	69.3 (224/323)	76.2	75.3	0.600
Dysplasia	96.1 (620/645)	84.9 (62/73)	81.6 (62/76)	83.2	81.2	
SCC	91.6 (591/645)	87.0 (141/162)	81.0 (141/174))	83.9	85.0	
Mean	88.7	85.5	77.3	81.1	80.5	
**NBI**	**Accuracy**	**Precision**	**Sensitivity**	**F1 Score**	**AP**	**Kappa**
Normal	88.0 (534/607)	87.6 (190/217)	79.2 (190/240)	88.0	84.5	0.653
Dysplasia	84.8 (515/607)	84.0 (136/162)	69.7 (136/195)	76.2	84.2	
SCC	92.1 (559/607)	87.4 (97/111)	80.8 (97/120)	84.0	86.7	
Mean	88.3	86.3	76.6	81.1	85.1	
**HSI**	**Accuracy**	**Precision**	**Sensitivity**	**F1 Score**	**AP**	**Kappa**
Normal	81.2 (500/616)	90.9 (230/253)	71.2 (230/323)	79.9	78.9	0.665
Dysplasia	97.7 (602/616)	89.7 (70/78)	92.1 (70/76)	90.9	83.6	
SCC	93.2 (574/616)	89.8 (149/166)	85.6 (149/174)	87.6	88.5	
Mean	90.7	90.1	83.0	86.1	83.7	

## Data Availability

The data presented in this study are available in this article.

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
