# Peer review of "Intelligent Identification of Early Esophageal Cancer by Band-Selective Hyperspectral Imaging"

_cancers, 2022, doi:10.3390/cancers14174292_

Round 1

Reviewer 1 Report

Thank your for submission. In your study, the hyperspectral imaging (HSI) technology and band selection were coupled with color reproduction, which was used for tumor detection. The manuscript was highly satisfied with the journal policy and the novelty was high. I recommend the submission without correction.

Reviewer 2 Report

In this study, the authors proposed a deep learning method to identify the esophageal cancer images into three categories. The paper is well written and can be published after addressing the following comments: 

1. The authors need to find out the limitations of the existing literature and need to point out the contribution of your work to overcome the limitations. 

2. In the conclusion, point out the limitation of your study and provide future research direction. 

3. Provide dataset description and availability in section 2. How you preprocess the dataset and gaining techniques is not clear. How did you use the dataset for training and so on? Provide an appropriate table. 

4. What kind of deep learning model is used? Compare the results with 4 to 5 recent state-of-the-art deep learning models. Which model performs better and why provide explanations?

5. Make the code available 

6. Cite recent works and classification techniques

i.  https://doi.org/10.3390/cancers14030554

ii. https://doi.org/10.3233%2FXST-200715

iii. https://doi.org/10.1016/j.semcancer.2022.01.007

iv.  https://doi.org/10.3390/ijerph19042013

v. https://doi.org/10.1016/j.compbiomed.2021.104649

Reviewer 3 Report

In this paper, the authors describe a new method for computer-assisted imaging and diagnosis of esophageal cancer. The authors described and compared the effectiveness of the combination of hyperspectral imaging (HSI) technology and band selection combined with color reproduction. At the same time, a new method based on machine learning was introduced to facilitate the diagnostic process. The new method showed a 5% improvement in accuracy compared to white-light imaging which is on par with NBI, so hyperspectral imaging (HSI) combined with AI may be a much better prospect for detecting esophageal cancer at early stages as well as a good and automated method for predicting the spectrum of this particular cancer. The study is well designed and described and I have no major substantive comments, but some minor flaws should be considered before publication.

Minor comments:

1.       Not every abbreviation is explained when it first appears in the text (mAP may seem obvious, but should be treated like any abbreviation in a manuscript)

2.       Minor language correction, for instance: “If the amount of data cannot meet the high-dimensional requirements, the accuracy be will reduced”. It should be “…, the accuracy will be reduced”.

3.       Figure 1. The presentation of the HSI algorithm should be improved in detail and resolution, as it is currently not readable, and I did not find it in the supplementary materials.

4.       The first 10 lines of the Results section belong to the Materials and Method section as a description of the validation methodology.

Round 2

Reviewer 2 Report

This paper can be accepted in this form.